# Non Coding RNAs as Regulators of Wnt/β-Catenin and Hippo Pathways in Arrhythmogenic Cardiomyopathy

**DOI:** 10.3390/biomedicines10102619

**Published:** 2022-10-18

**Authors:** Marina Piquer-Gil, Sofía Domenech-Dauder, Marta Sepúlveda-Gómez, Carla Machí-Camacho, Aitana Braza-Boïls, Esther Zorio

**Affiliations:** 1Unit of Inherited Cardiomyopathies and Sudden Death (CaFaMuSMe), Health Research Institute La Fe, 46026 Valencia, Spain; 2Center for Biomedical Network Research on Cardiovascular Diseases (CIBERCV), 28015 Madrid, Spain; 3Cardiology Department, Hospital Universitario y Politécnico La Fe, 46026 Valencia, Spain

**Keywords:** arrhythmogenic cardiomyopathy, ncRNA, lncRNA, miRNA, circRNA, pseudogene, ceRNA, Wnt/β-catenin pathway, Hippo pathway

## Abstract

Arrhythmogenic cardiomyopathy (ACM) is an inherited cardiomyopathy histologically characterized by the replacement of myocardium by fibrofatty infiltration, cardiomyocyte loss, and inflammation. ACM has been defined as a desmosomal disease because most of the mutations causing the disease are located in genes encoding desmosomal proteins. Interestingly, the instable structures of these intercellular junctions in this disease are closely related to a perturbed Wnt/β-catenin pathway. Imbalance in the Wnt/β-catenin signaling and also in the crosslinked Hippo pathway leads to the transcription of proadipogenic and profibrotic genes. Aiming to shed light on the mechanisms by which Wnt/β-catenin and Hippo pathways modulate the progression of the pathological ACM phenotype, the study of non-coding RNAs (ncRNAs) has emerged as a potential source of actionable targets. ncRNAs comprise a wide range of RNA species (short, large, linear, circular) which are able to finely tune gene expression and determine the final phenotype. Some share recognition sites, thus referred to as competing endogenous RNAs (ceRNAs), and ensure a coordinating action. Recent cancer research studies regarding the key role of ceRNAs in Wnt/β-catenin and Hippo pathways modulation pave the way to better understanding the molecular mechanisms underlying ACM.

## 1. Arrhythmogenic Cardiomyopathy

Arrhythmogenic cardiomyopathy (ACM) is a rare disease with an estimated prevalence between 1:1000 and 1:5000 (OMIM #107970; ORPHA247) characterized by fibrofatty replacement of the ventricular myocardium and a high ventricular arrhythmia burden with increased risk of sudden cardiac death [1,2,3,4,5,6], sometimes as the first manifestation of the disease. ACM has been defined as a desmosomal disease caused by mutations in any of the five genes encoding desmosomal proteins: desmocollin-2 (*DSC2*), desmoglein-2 (*DSG2*), desmoplakin (*DSP*), plakoglobin (*JUP*), and plakophilin-2 (*PKP2*). Mutations in these genes are responsible of more than 60% of genotyped ACM cases. However, mutations in 14 non-desmosomal genes have also been identified in 5% of patients, namely in α-actinin-2 (*ACTN2*), cadherin-2 (*CDH2*), αT-catenin (*CTNNA3*), desmin (*DES*), filamin C (*FLNC*), LIM domain binding protein 20 (*LDB3*), lamin A/C (*LMNA*), transmembrane protein 43 (*TMEM43*), transforming growth factor beta 3 (*TGFB3*), tight junction protein 1(*TJP1*), titin (*TTN)*, RNA-binding motif protein 20 (*RBM20*), sodium voltage-gated channel, alpha subunit 5 (*SCN5A*), and phospholamban (*PLN*) (Table 1) [1,3,4,5,6,7]. Historically, ACM caused by mutations in desmosomal genes was associated with isolated or predominantly right ventricular disease (RV-ACM). However, recent studies have highlighted a high prevalence of left ventricular involvement along with isolated and left-dominant ACM forms (LV-ACM) in patients carrying *FLNC*, *DSP*, or *DSG2* mutations, for instance [3,8,9,10]. 

ACM is characterized by the fibrofatty replacement of the myocardium, typically progressing from the epicardium to the endocardium, and transmural fibrofatty infiltration can also be observed in advanced stages of the disease [1,4,5,6,10,18]. 

The electrical heterogeneity created by these pathological changes generates circuits for re-entrant ventricular tachyarrhythmias. Thus, the hallmarks of ACM pathogenesis include cardiomyocyte loss, fibrosis, adipogenesis, inflammation, and arrhythmogenesis [4]. All these cellular mechanisms are interrelated and are orchestrated by the same signaling pathways, which in turn are also connected. 

Desmosomes are crucial for myocardium integrity and function, because they ensure cell-to-cell communication as well as intracellular signaling through the Wnt/β-catenin and Hippo pathways [18]. 

## 2. Dysregulated Signaling Pathways in ACM

The inhibition of the canonical Wnt/β-catenin pathway and the activation of the Hippo and TFGβ pathways appear to play key roles in ACM physiopathology by regulating the adipogenic and fibrotic cascades [4,18,19,20,21,22]. 

### 2.1. Wnt/β-Catenin Pathway in ACM

The Wnt/β-catenin pathway orchestrates heart development and regulates cardiac tissue homeostasis in adults. 

Briefly, the activation of the membrane Frizzled receptor (Fz) by WNT ligands triggers the recruitment of low-density lipoprotein receptor-related proteins 5 and 6 (LRP5/6) and Dishevelled (DVL). This macromolecular complex also assembles cytoplasmic proteins, such as Axin and glycogen synthase kinase 3β (GSK3β), and releases free β-catenin in the cytoplasm. In its unphosphorylated form, free β-catenin escapes destruction and translocates into the nucleus. Once there, it binds the transcription factors TCF/LEF to activate the transcription of genes responsible for adipogenesis, fibrosis, cell cycle, and cell differentiation (Figure 1a). In contrast, in the absence of WNT ligands, Fz and LRP5/6 receptors remain uncoupled at the membrane, whereas Axin and GSK3β build the degradation complex in the cytoplasm, together with adenomatous polyposis coli (APC) and casein kinase 1 (CK1). This protein complex allows GSK3β to phosphorylate β-catenin. Phosphorylation marks on β-catenin are detected by the ubiquitin ligase beta-transducin repeat containing E3 (β-TrCP), and trigger its ubiquitination and subsequent degradation (Figure 1b), so that no profibrotic or proadipogenic genes are expressed [4,18,19,21,23].

The canonical Wnt signaling pathway has been fully accepted as partially responsible for ACM pathogenesis [4,18,19,21,23,24,25] according to the “dysregulated signalling hypothesis” postulated by Prof Marian et al. [26], and further supported by the structural resemblance between β-catenin and plakoglobin (also known as γ-catenin). They demonstrated that the desmosomal plakoglobin translocated into the nucleus in an in vitro ACM model, finally leading a shift in the expression of Wnt-dependent genes and an increase in intracellular fat droplets [19]. The altered integrity of the desmosomes in ACM cases may increase the pool of cytoplasmatic plakoglobin, which favors its nuclear translocation and slow β-catenin degradation at the proteosome. Nuclear plakoglobin interacts with transcription factors, with the final effect of an unbalanced Wnt/β-catenin pathway yielding an increased transcription of proadipogenic and profibrotic genes [27]. In recent years, direct and indirect evidence for suppressed Wnt signaling has been gathered from several cellular and/or animal models of ACM-causing mutations in *JUP*, *PKP2*, *DSG*, and *DSP* genes (Table 2) [13,16,19,24,25,28,29]. 

### 2.2. Hippo Pathway in ACM

Closely linked to the Wnt/β-catenin pathway, the Hippo pathway has also been regarded [24] as an important regulator of adipogenesis, cell proliferation and differentiation. Yes-associated protein (YAP) and transcriptional coactivator with PDZ-binding motif (TAZ) represent the core of this pathway, which is controlled by two Ser/Thr kinases, namely Ste20-like kinase1/2 (MST1/2) and large tumor suppressor kinase 1/2 (LATS1/2), and their cofactors Salvador1 (SAV1) and monopolar spindle-one-binder protein (MOB1) [30,31]. MST1/2 and its cofactor SAV1 phosphorylate LATS1/2 and MOB1, which in turn phosphorylate YAP and TAZ ensuring their degradation in the cytoplasm. Moreover, the activity of MST1/2 and LATS1/2 is regulated by neurofibromin 2 (NF2) which, in turn, could be inactivated by the phosphorylated form of PKCα [32]. Briefly, upon YAP Ser127 and Ser381 phosphorylation, the ubiquitination and degradation of YAP is promoted through its interaction with CK1 and the ubiquitin ligase β-TrCP [30,31] (Figure 2a). In contrast, if the Hippo pathway is not activated, YAP/TAZ complex accumulates in the cytoplasm instead of being degraded, allowing YAP/TAZ to translocate into the nucleus where it binds to the transcription factors TEAD/TEF or SMAD to promote the expression of proliferation and differentiation genes (Figure 2b).

In ACM pathogenesis, the abnormal cell–cell interaction unbalances the Hippo pathway, increasing the activity of the previously mentioned kinases [33]. Cell remodelling and mechanotransduction signals further promote the phosphorylation of YAP/TAZ by the ACM-related reduction PKCα in the membrane, and the subsequent higher level of NF2 activity [15]. All these steps result in phosphorylated YAP/TAZ cytoplasmatic sequestration and inhibition of the expression of the Hippo target genes. Several authors have studied the crosstalk between the Wnt/β-catenin and Hippo pathways. Although not all the molecular mechanisms are fully understood, the recognized effect in ACM can be summarized as reduced Wnt/β-catenin and Hippo/YAP signaling with a subsequent increase in proadipogenic gene expression [4,15,18,21,34]. Firstly, as previously mentioned, cytoplasmic PG competes with β-catenin to bind TCF/LEF, although in a weaker manner, with a net effect of suppressing of the Wnt/β-catenin pathway and inducing a shift from a myocardiogenic to an adipogenic phenotype, by reducing the expression of cyclin-D1 and c-Myc and enhancing the transcription of proadipogenic and apoptotic genes [13,19,29] (Figure 3a). Secondly, experiments with coimmunoprecipitation demonstrated that phospho-YAP binds phospho-β-catenin (Figure 3b) or PG (Figure 3c) in ACM, although the functional significance of these interactions remains unclear [15]. Finally, both pathways can also be interconnected by TAZ activity. Varelas et al. demonstrated that phospho-TAZ binds and blocks Dvl, negatively regulating the Wnt/β-catenin pathway [35] (Figure 3d).

## 3. Non-Coding RNAs

Notwithstanding that 80% of DNA is transcribed, less than 2% encodes for proteins. There is an increasing interest in unravelling the roles of the remaining 98% of the genome, previously named junk DNA (including introns and spacer DNA). The potential role as regulator of a wide range of biological functions is well established for at least a fraction of this junk DNA, i.e., the well-named non-coding RNAs (ncRNAs). Indeed, ncRNAs are known to control cellular behavior through modulation of gene expression at a transcriptional or post-transcriptional level. Moreover, ncRNAs orchestrate intracellular pathways in a fine-tuned manner due to their ability to bind proteins and/or other RNA species [36,37,38,39].

Although ncRNAs do not encode for proteins, they can be transcribed, either constitutively or under regulation (Figure 4). The housekeeping ncRNAs comprise several species, including small nucleolar RNAs (snoRNA), ribosomal RNA (rRNA), or transfer RNA (tRNA). snoRNAs (60–300 bp), usually encoded in intronic regions, are among the small nucleolar riboproteins responsible for rRNA modifications. rRNA represents 80% of total cellular RNA and together with tRNAs makes possible protein translation [36,38,40,41]. 

Meanwhile, the regulatory ncRNAs are heterogeneous in size and form, including small microRNAs (miRNAs) or piwi-interacting RNAs (piRNAs), and larger long non-coding RNAs (lncRNAs), circular RNAs (circRNAs), or pseudogenes. Their characteristics and functions are detailed in the following. 

In recent years, a huge effort has been made by the scientific community to study the effects of individual miRNAs on their target mRNAs, aiming to improve knowledge of the complex mechanisms governing these interactions. However, the paradigm has changed, and recent findings supporting the multiple and interconnected functions of different ncRNAs species within the networks called competing endogenous RNAs (ceRNAs) have shed light on the molecular mechanisms underlying certain complex diseases. ACM is a far from simple disease and thus, ncRNAs may present a promising opportunity to harmonize the complex interactions underlying the Wnt/β-catenin and Hippo pathways.

Salmena et al. postulated the hypothesis by which ncRNAs interact among themselves, forming a regulatory network focused on communication mediated by MicroRNA response elements (MREs) called ceRNAs (Figure 5) [42]. The canonical role of these MicroRNA response elements (MREs) is to bind miRNAs and to interfere with the translation of their target mRNAs (Figure 5a,b). However, the complexity of their mechanisms of action required the definition of the new concept of ceRNAs, to better understand the interaction among different types of ncRNAs. For example, lncRNAs (including circRNAs) may behave like sponges for miRNAs, which then become unable to bind their target mRNAs so that the protein translation remains preserved (Figure 5c). The second model of ceRNAs would apply in situations where miRNAs lose their inhibitory action on their target mRNAs because the MREs present in the 3′UTR are occupied and blocked by lncRNAs (Figure 5d). Finally, ceRNAs can include pseudogenes that present similar MREs to their original genes and consequently are able to bind their target miRNAs, hampering their inhibitory effect on the translation of the original gene (Figure 5e).

This plethora of functions has placed ncRNAs in the research spotlight, as promising key molecules to increase knowledge and understanding of biological and pathophysiological processes and to pave the way for design of new therapeutic strategies. 

### 3.1. Biogenesis and Function of microRNAs (miRNAs)

The best-studied kind of ncRNA are the small single-strand sequences (20–22 nt) named miRNAs [43].

They can be encoded in intronic regions of their target mRNAs or in intergenic positions. miRNAs are transcribed into the nucleus as pri-miRNAs (more than 1000 nt) by the enzymes RNA polymerase II and III. Still in the nucleus, Drosha processes them into 60–120 nt pre-miRNAs. This stem-loop RNA is exported to the cytoplasm by exportin 5 and Ran-GTPase. Then, the pre-miRNA is processed by Dicer into a short double-strand molecule, which is recruited by the RISC complex in order to select one of the strands to interact with Argonaute proteins (Ago) and finally exert its function by targeting the 3′UTR of its target mRNA [36,37,38,44,45] (Figure 6).

miRNAs can bind their complementary MRE in the 3′UTR of their target mRNAs to inhibit gene translation. However, they can also bind MREs in the 5′UTR or even in coding regions, although less frequently. If the seed sequence and the MRE bind with perfect complementarity, then the mRNA molecule’s fate is degradation. However, if the complementarity is imperfect, which is more common, then the translation of the target mRNA is inhibited [37,38,44,45]. 

Interestingly, their expression is cell-specific and their final effect is complex because they can regulate different pathways in a coordinated manner by blocking the translation of certain mRNAs at different levels of the same signaling pathway [45,46]. Furthermore, several miRNAs can cooperate to target a given mRNA, and a single miRNA can target several lncRNAs and different mRNAs from up- or downstream in the same pathway, thus acting as master regulators of gene transcription [36,38,47].

Many studies have addressed the role of miRNAs in pathological cardiac processes such as fibrosis after myocardium infarction (MI), hypertrophy, atrial fibrillation, or heart failure [48,49,50,51,52,53,54,55,56,57,58]. Regarding ACM, very few studies have been published. Those that have were focused on different samples and employed different technologies, making it difficult to draw conclusions about the effect of miRNAs on ACM pathogenesis. Remarkably, none of the previous studies linked miRNAs with the Wnt/β-catenin or Hippo pathways. Firstly, Rainer et al. explored miRNA differences by using Taqman low-density arrays (768 miRNAs included) on cardiac stem cells isolated from three biopsies from ACM patients and three from explanted hearts, yielding three miRNAs with significant differences in their expression (miR-520c-3p, miR-29b-3p, and miR-1183) [59]. Secondly, Bueno Marinas studied nine ACM cases and four controls (RV tissue and plasma) by means of an 84-miRNA cardiac-related array. After validation in a larger cohort of samples, they concluded that a group of six miRNAs (miR-122-5p, miR-133a-3p, miR-133b, miR-142-3p, miR-182-5p, and miR-183-5p) presented high discriminatory diagnostic power in ACM patients [60]. Thirdly, Sacchetto et al. screened 754 miRNAs in 21 pooled plasmas from ACM patients and 20 controls. They found five differently expressed miRNAs (miR-20b, miR-505, miR-250c, miR-590-5p, and miR-185-5p) [61]. Finally, Khudiakov et al. focused their study on the miRNA composition of pericardial fluid from six ACM patients and three controls by using small RNA sequencing. They identified five miRNAs (miR-1-3p, has-miR-21-5p, miR-122-5p, miR-206, and miR-3679-5p) with different levels between the two clinical groups [56]. 

### 3.2. Biogenesis and Functions of PIWI-Interacting RNAs (piRNA)

piRNAs are small ncRNAs (24–30 nt) that also regulate gene expression but without the participation of DICER in their biogenesis. They can be encoded in transposable elements in either non-coding or coding gene regions. RNAPol II is responsible for producing long single-stranded RNA precursors which are matured into fragments of 25–40 nt by the endonuclease MitoPLD/Zucchini (Zuc). The final length of the mature piRNA depends on the target PIWI proteins to which the piRNA binds [38,62]. 

The wide range of ways by which piRNAs can regulate gene expression suggests that these piRNAs are more powerful regulators than the better-known miRNAs. The mechanisms of action of these ncRNAs require the association of ncRNA to the PIWI proteins to suppress expression of transposable elements, and their transcription could be regulated by DNA methylation or histone modification [62]. 

### 3.3. Biogenesis and Functions of Long Non-Coding RNAs (lncRNA)

lncRNAs are the longest ncRNAs, representing a heterogeneous group characterized just by their length (more than 200 nt). There are many differences among them, according to the location in which they are encoded, post-transcriptional modifications, binding properties, final function, etc. (Figure 4). Similarly to protein-coding DNA, the DNA fragments that encode these lncRNAs contain multiple exons (although less than protein-coding transcripts). When the RNA Pol II transcribes the sequence, the resulting RNA undergoes splicing and usually includes a 5′ cap and 3′ poly(A) tail [36,40,63,64]. The expression of lncRNAs can be activated or inhibited by epigenetic mechanisms. Attending to their location within the genome they can be identified with their own nomenclature as enhancers, antisense, intronic, or intergenic, also known as long intergenic non-protein coding RNA (LINCRNA) [36,38,65]. 

lncRNAs can exert local effects on sites close to their transcription sites, known as cis effects. Thus, they are able to regulate the expression of neighboring genes at an epigenetic, transcriptional, or post-transcriptional level. Due to their ability to bind proteins (e.g., transcription factors, chromatin/DNA modifying enzymes, or RNA-binding protein (RBPs)) or other nucleic acids (DNA, mRNA, or ncRNAs), they act as scaffold ensuring the proximity among all payers in this complex regulation. Additionally, they can also act at distant genomic or cellular locations (trans effect) as cell-to-cell communicators [40]. 

Many different functions have been attributed to lncRNAs (Figure 7) [40,47,65]. Firstly, lncRNAs can regulate gene expression by controlling the recruitment of essential proteins for transcription regulation (e.g., methyltransferases, demethylases, acetyltransferases, or deacetylases) (Figure 7a,b). Secondly, lncRNAs can interact with other RNA species such as mRNAs, regulating their splicing or stability and, as a consequence, their translation into proteins (Figure 7c,d). Finally, the presence of repeated MREs in lncRNAs allows them to act as sponges for miRNAs by hijacking them (and subsequently inactivating them) or by blocking the binding sites in the 3′UTR of their target mRNAs (Figure 7e) [36,38,40,65]. 

An example of a well-known ceRNA orchestrated by a lncRNA is the overexpression of MIAT (myocardial infarction-associated transcript) in myocardial infarction. This lncRNA can sponge the negative regulator miR-24, and subsequently induce an overexpression of miR-24 target mRNAs, Furin and TGF-β1, in cardiac fibroblasts [66]. Another example is the ceRNA coordinated by the lncRNA HOX transcript antisense RNA (HOTAIR), which acts as a sponge for miR-613, negatively regulating the expression of connexin-43 in atrial fibrillation [67]. 

#### 3.3.1. Biogenesis and Functions of Circular RNAs (circRNAs)

circRNAs are a kind of covalently closed single-stranded lncRNAs generated by reverse splicing of mRNA precursors (pre-mRNAs), which means that the 5′ splice site is directly joined with a downstream 3′ splice site to be covalently bonded [44,68,69,70,71,72]. Notably, circRNAs can be hosted in coding genes whose levels are in turn regulated by those circRNAs. This is the case for the most abundant circRNA in the heart, the circSLC8A1, a single-exon circRNA codified in exon 2 of the SLC8A1 gene which encodes the Na^+^/Ca^++^ exchanger [63,73]. It is also worth noting the characterization of 402 circRNAs contained within the huge TTN gene and containing from 1 to 153 exons [63]. 

Interestingly, circRNA expression is mostly tissue-specific due to the highly cell-type-specific manner of interaction with RBP [68]. Due to their special circular conformation, lacking 3′ poly(A) tail and 5′ cap structures, circRNAs are more resistant to degradation by exonucleases than linear sequences. Consequently, they have a longer half-life suitable for measurement in biofluids [44,68,74].

The most studied mechanism of action of circRNAs is their role as miRNA sponges. This feature stems from their ability to present repeated MREs, acting as effective sponges of miRNAs, or even blocking their complementary sequences in the 3′UTR of the target mRNAs (Figure 7) [40,47,63,68,74]. In addition to behaving as a miRNA sponge, circRNAs can bind proteins and so act as a scaffold to promote protein–protein interactions, or can sponge target proteins to modify their fate [40,47,65,68,74]. Two circRNAs widely expressed in the heart are heart-related circRNA (HRCR) and CDR1. Both act as miRNA sponges; HRCR was observed to bind miR-223 in a murine model of cardiac hypertrophy and heart failure [75], while CDR1 mediates post-myocardial infarction damage in mice by sponging miR-7 [76]. 

Finally, as an exception to the pivotal definition of ncRNAs, cases have been described in which circRNAs can be translated into proteins. This is the case with CircAXIN1, which encodes for an axin-like protein named AXIN1-2955aa that promotes gastric cancer development through Wnt/β-catenin pathway destabilization [77].

#### 3.3.2. Pseudogenes, Biogenesis and Functions

The pseudogenes represent another kind of lncRNA, including similar sequences to those of coding genes but lacking the possibility of being translated. GENECODE v.31 identified 13,000 pseudogenes [78], although they could be considered relics of evolution or unprofitable DNA. However, although these sequences resemble the original gene, they usually harbor premature stop codons, deletions, insertions or even frameshift mutations that impair their translation into proteins [78,79,80]. These are the non-processed pseudogenes. In contrast, pseudogenes can also result from processed genetic elements such as the insertion of the reverse transcription of the original mRNA (retrotransposition) [80,81]. 

The transcription of pseudogenes is regulated by proximal regulatory elements due to the lack of canonical promoters, and their expression seems to be tissue-dependent and associated with pathologic conditions (e.g., cancer) [40,81].

Their similar sequence to the original genes allows them to act as ceRNAs by modulating the effect of other ncRNAs on the expression of their target genes [40,47,65]. PTEN and its pseudogene PTENP1 have been widely studied because of their implication in cancer pathogenesis. The main difference between PTENP1 and PTEN results from the lack of an initiation codon due to a missense mutation impairing its translation, and the 3′UTR of the pseudogene is 1 kb shorter than in the original sequence. Despite the shorter length of the 3′UTR, PTENP1 conserves MREs to bind miR-17, miR-21, miR-214, miR-19, and miR-26. Thus, PTENP1 3′UTR functions as a sponge for PTEN-targeting miRNAs [81]. For this reason, copy number variations in pseudogenes can strongly interfere with the pathogenesis of many diseases. Indeed, it has been reported that an increase in the copy number of the pseudogene NOTCH2NL is associated with autism, whereas a reduction in the copy number of this pseudogene is related to schizophrenia [82].

Study of the role of ncRNAs in pathophysiology requires a multifactorial approach to assess the degree of implication of each player in these complex scenarios orchestrated by ceRNAs.

## 4. Non-Coding RNAs Databases

The scientific community has devoted huge efforts to better understanding and classifying ncRNA species and their potential interconnections. Recent developments have rendered previous approaches too simple and obsolete. It is no longer valid to explain the molecular pathogenesis of a complex disease only based upon the effect of a single miRNA on its target mRNAs, as the first studies on the topic did. Nowadays, the study of ncRNAs requires an open-minded approach in which proteins, RBP, nucleic acids, and ncRNAs act within interconnected ceRNA networks.

In order to update the published information regarding the validated connections among the aforementioned ncRNAs, many online databases have been created (Table 3).

## 5. Non-Coding RNAs in ACM

Only a few studies have focused on the role of ncRNAs in ACM, and all have restricted their results to one species, namely miRNAs, concluding that they emerge as potentially useful biomarkers [51,52,53,54,55,56,57,59,61,92,93]. To the best of our knowledge, no studies have addressed the regulation of the Wnt/Hippo pathway by ncRNAs in the scenario of ACM. However, due to the importance of both pathways in the pathogenesis of many cancers, other studies have shed light on the issue (Table 4). 

### 5.1. Non-Coding RNAs in the Wnt/β-Catenin Pathway 

Recent research on osteosarcoma has suggested that lncRNA MRLP23-AS1 might be overexpressed and act as a sponge for miR-30b, so that the miR-30b inhibitory effect on its target mRNA (myosin heavy chain 9, MYH9) is reduced to provoke an increase of the final level of this protein. Meanwhile, MYH9 is a trans-activator of the expression of β-catenin, thus exerting a final activating effect on the Wnt/β-catenin pathway [98]. Moreover, β-catenin is also indirectly regulated by lncRNA LSINCT5. This lncRNA has been shown to be upregulated in breast cancer and is capable of sponging miR-30a, which inhibits β-catenin translation [100]. DANCR (differentiation antagonizing non-protein coding RNA) is one of the better studied lncRNAs, because of its implication in the regulation of the Wnt/β-catenin signalling pathway [102,110]. The lncRNA DANCR regulates β-catenin translation not only indirectly by sponging miR-216a [104], miR-214, or miR-320a [102,103] and modulating the inhibitory effect of these miRNAs on the mRNA of β-catenin, but also directly by binding β-catenin mRNA so that MREs are blocked and there is no inhibitory effect on translation [102,103]. 

Yang et al. went further in their study in which they corroborated that lncRNA long intergenic non-protein coding RNA 467 (LINC00467) was upregulated in lung adenocarcinoma mediated by STAT1. Moreover, LINC00467 was responsible for the downregulation of the Wnt/β-catenin pathway inhibitor, DKK1, by recruiting the histone methyltransferase EZH2 and epigenetically silencing its expression downstream. In this way, by reduction of the inhibitory effect of DKK, the Wnt/β-catenin pathway was overactivated in lung adenocarcinoma [101].

### 5.2. Non-Coding RNAs in the Hippo Pathway

The Hippo pathway has also been widely studied in the pathogenesis of several cancers, because of its crosslink with the Wnt-β-catenin pathway. Several studies have demonstrated that certain miRNAs are responsible for the regulation of YAP, such as miR-550a-3-5p [111], -27b-3p [112], -195-5p [113], or TAZ by miR-9-3p [114]. 

Additionally, the Hippo pathway might be also regulated indirectly. The translation of the kinase responsible for YAP phosphorylation, LATS2, is regulated by miR-103a-3p and -429 in colorectal cancer, so YAP phosphorylation is reduced and subsequently its nuclear localization increases [115]. Probably, as already mentioned, the regulation of the final levels of these miRNAs are further regulated at a much higher level of complexity by ceRNAs that accurately determine the finely tuned regulation of this pathway. In this respect, the function of lncRNA FLVCR1-AS1 seems to regulate the Hippo pathway in ovarian serous cancer by sponging a negative regulator of the translation of YAP, the miR-513 [106]. 

Interestingly, YAP/TAZ can also modulate the transcription of other genes by binding proteins or even their mRNAs. In gastric cancer, regulation of the expression of YAP1 and lncRNA RP11-323N12.5 seems to be bidirectional. The expression of RP11-323N12.5 seems to be regulated by YAP/TAZ/TEAD, while RP11-323N12.5 activates YAP1 transcription by acting as scaffold and promoting the binding of the transcription factor c-MYC to the YAP1 promoter [107]. However, its circular RNA (circYap) negatively regulates YAP translation due to its ability to bind YAP mRNA and the translation initiation proteins (eIF4G and PABP), as has been described in breast cancer where overexpression of circYap reduces YAP translation by suppressing the formation of transcriptional initiation machinery without modifying YAP mRNA levels [108]. 

## 6. Concluding Remarks

From a clinical standpoint, the extraordinary complexity of ACM is supported by the increasing list of causal genes, lists of major and minor diagnostic criteria, the discussion about the criteria to be used (Task Force of Padua), and the wide range of phenotypic features involving the right ventricle, the left ventricle, or both, with or without relevant inflammation or even with associated non-compaction. From a molecular perspective, the complexity grows exponentially in terms of interwoven pathways underlying the well-known fibrofatty replacement of the ventricular myocardium, of which the most widely accepted are the Wnt/β-catenin and Hippo pathways. The ultimate mechanisms orchestrating all the necessary elements to produce ACM remain broadly unknown. A few studies have reported scarce results regarding measurement of miRNAs in ACM, and none of them linked the singular differences with molecular pathways. However, the other ncRNAs have not yet been addressed in ACM. Taking advantage of great efforts made in cancer research, we have presented in this paper evidence of the role of ncRNAs in modulating the Wnt/β-catenin and Hippo pathways. A challenge is thus presented. Can the presented interactions or others in the same scenario help to improve understanding of ACM pathophysiology? Will revealing ceRNA networks involved in ACM be able to open avenues leading to the development of new therapeutic options for patients? A new era of research has just begun.

## Figures and Tables

**Figure 1 biomedicines-10-02619-f001:**
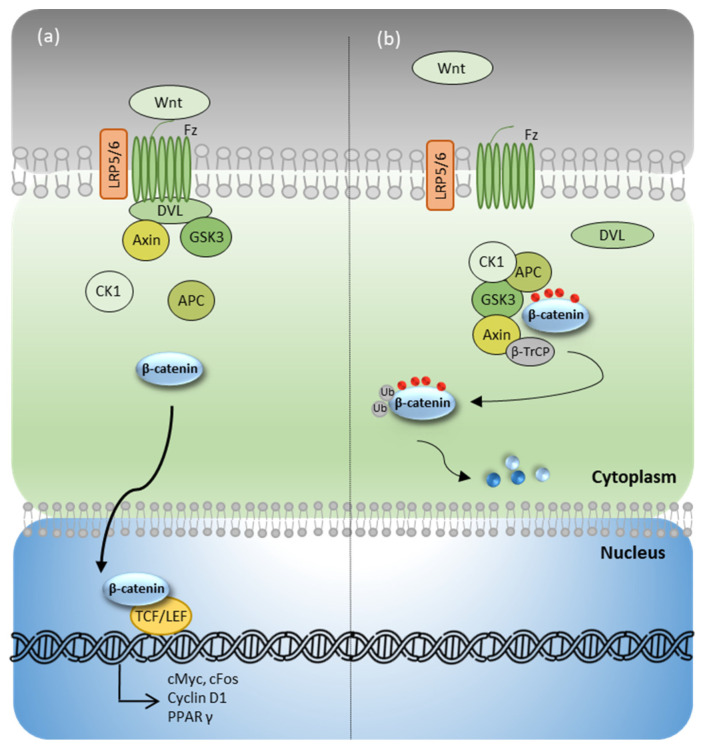
Wnt/β-catenin pathway. (**a**) Wnt/β-catenin pathway in the presence of Wnt ligand. (**b**) Wnt/β-catenin pathway in the absence of Wnt ligand. APC: adenomatous polyposis coli; β-TrCP: beta-transducin repeat containing E3 ubiquitin protein ligase; CK1: casein kinase 1; DVL: Dishevelled; Fz: Frizzled receptor; GSK3β: glycogen synthase kinase 3β; LRP5/6: low-density lipoprotein receptor-related proteins 5 and 6; PPARγ: peroxisome proliferator-activated receptor gamma; TCF/LEF: T cell factor/lymphoid enhancer factor family.

**Figure 2 biomedicines-10-02619-f002:**
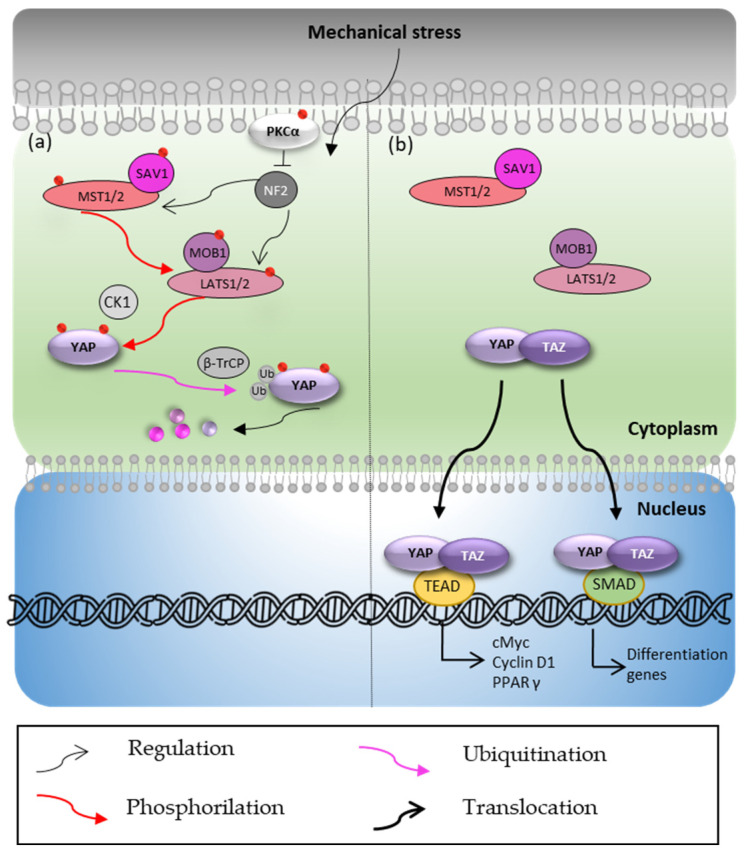
Hippo pathway. (**a**) Hippo pathway in the presence of stimulus. (**b**) Hippo pathway in the absence of stimulus. CK1: casein kinase; β-TrCP: beta-transducin repeat containing E3 ubiquitin protein ligase; SAV1: Salvador1; MST1/2: Ste20-like kinase 1/2; MOB1: monopolar spindle-one-binder protein 1; LATS1/2: large tumor suppressor kinase 1/2; SMAD: suppressor of mothers against decapentaplegic; TAZ: transcriptional coactivator with PDZ-binding motif; TEAD: transcriptional enhanced associate domain Ub: ubiquitination marks; YAP: Yes-associated protein. Red points indicate phosphorylation sites.

**Figure 3 biomedicines-10-02619-f003:**
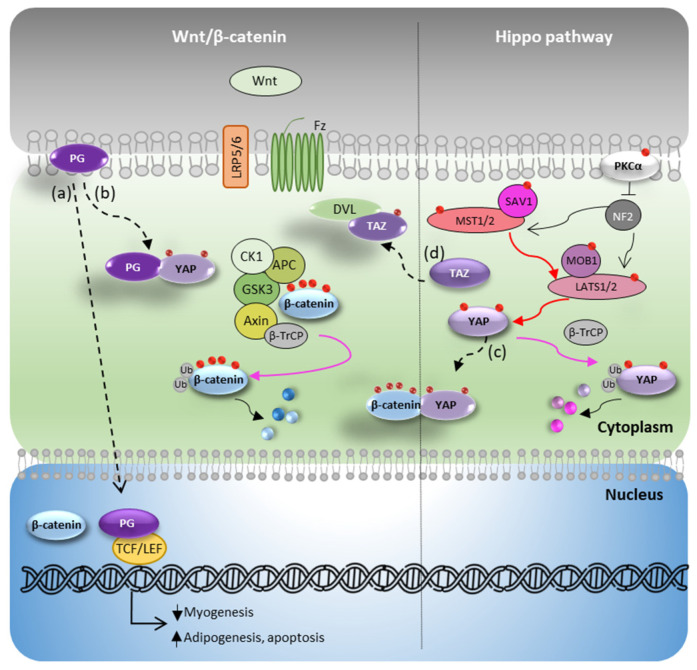
Interconnection between Wnt/β-catenin and Hippo pathways in the pathologic ACM scenario. (**a**) Free plakoglobin translocates into the nucleus and competes with β-catenin to bind the transcription factors TFC/LEF, although in a weaker manner, thus producing a shift from myogenic to adipogenic cell phenotype; (**b**) Free plakoglobin binds and blocks phospho-YAP, impairing its degradation; (**c**) phospho-YAP binds phospho-β-catenin, avoiding its ubiquitination; (**d**) phospho-TAZ binds Dvl, blocks β-catenin translocation to the nucleus, and impairs gene transcription. APC: adenomatous polyposis coli; β-TrCP: beta-transducin repeat containing E3 ubiquitin protein ligase; CK1: casein kinase 1; GSK3β: glycogen synthase kinase-3 beta; SAV1: Salvador1; MST1/2: Ste20-like kinase 1/2; MOB1: monopolar spindle-one-binder protein 1; PG: Plakoglobin; LATS1/2: large tumor suppressor kinase 1/2; NF2: neurofibromatosis type 2; TAZ: transcriptional coactivator with PDZ-binding motif; Ub: ubiquitination marks; YAP: Yes-associated protein. Discontinuous arrows indicate pathological ACM situations.

**Figure 4 biomedicines-10-02619-f004:**
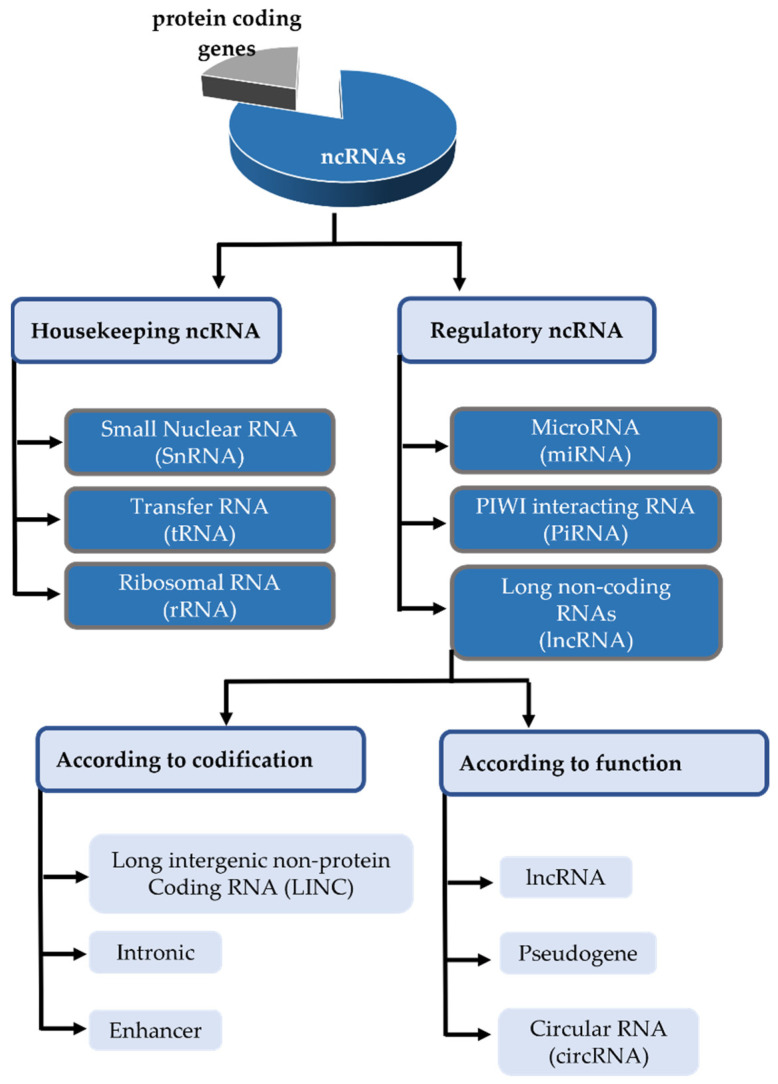
Classification of the non-coding transcriptome.

**Figure 5 biomedicines-10-02619-f005:**
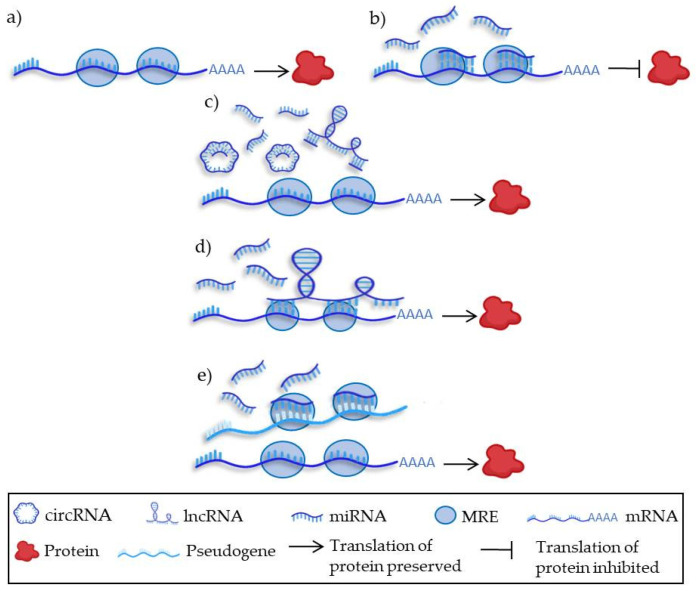
Main models of competing endogenous RNA (ceRNAs). (**a**) mRNAs are rich in MREs; (**b**) miRNAs bind MREs in the 3′UTR of their target mRNAs and inhibit protein translation; (**c**) ncRNAs such as lncRNAs or circRNAs are rich in MREs and can gather miRNAs, reducing the MiRNAs’ inhibitory effect on the translation. (**d**) lncRNAs can block the MREs in mRNAs, impairing the binding of miRNAs and reducing the negative influence on protein translation. (**e**) Pseudogenes can also bind miRNAs, avoiding their effect on the original gene and allowing their translation. circRNA: circular RNA; lncRNA: long non-coding RNA; miRNA: microRNA; MRE: MicroRNA response element; mRNA: messenger RNA.

**Figure 6 biomedicines-10-02619-f006:**
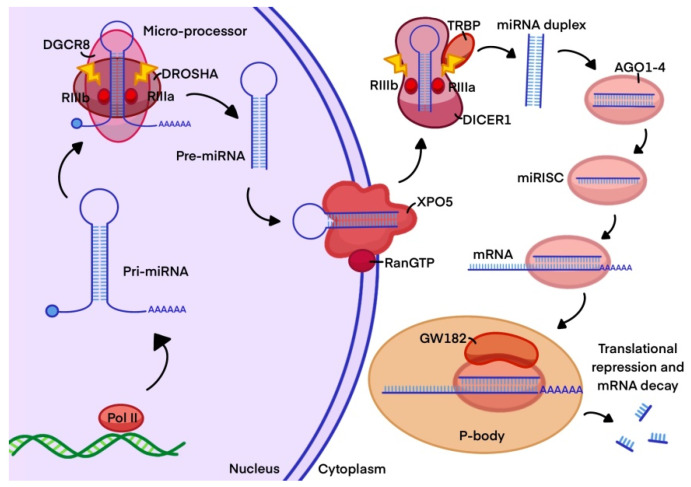
Biogenesis of miRNAs. miRNAs are transcribed as hairpin pri-miRNAs (1000 nt) by the RNA polymerases II and III. Drosha processes pri-miRNAs into shorter pre-miRNAs (60–120 nt). This stem-loop RNA is exported to the cytoplasm by exportin 5 and Ran-GTPase. Then, the pre-miRNA is processed by Dicer into a short double-strand RNA molecule, which is recruited by the RISC complex to select one of the strands to interact with the protein Argonaute (Ago), to exert its function by targeting the 3′UTR target mRNA.

**Figure 7 biomedicines-10-02619-f007:**
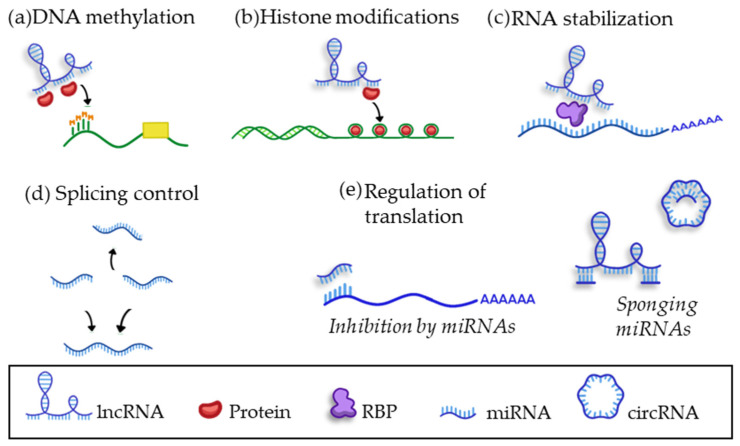
Main mechanisms of action of lncRNAs. (**a**–**c**) lncRNAs bind key proteins to promote epigenetic mechanisms such as (**a**) DNA methylatyion if lncRNA binds DNA transferases; (**b**) Histone acetylation if it binds histone acetylases/deacetylases, or (**c**) stabilizing mRNA by binding riboproteins. (**d**,**e**) lncRNA can bind other types of ncRNAs to regulate (**d**) alternative splicing, or (**e**) by blocking MREs in mRNAs or sponging miRNAs. lncRNA: long non-coding RNA; miRNA: microRNA; RBP: RNA binding protein.

**Table 1 biomedicines-10-02619-t001:** List of genes associated with arrhythmogenic cardiomyopathy.

Cell Structure	Gene	Protein	Chromosomal Location	Anatomic Affection	Related to Wnt/β-Catenin
**Desmosome**	*DSC2*	Desmocollin-2	18q12.1	RV/BiV	
	*DSG2*	Desmoglein-2	18q12.1	RV/LV/BiV	
	*DSP*	Desmoplakin	6p24.3	LV/BiV	[11,12]
	*JUP*	Plakoglobin	17q21.2	RV/BiV	[13]
	*PKP2*	Plakophilin-2	12p11.21	RV/BiV	[14,15,16]
**Cytoskeleton**	*ACTN2*	α-actinin-2	1q43	LV	[17]
	*CDH2*	Cadherin-2	18q12.1	RV/BiV	
	*CTNNA3*	Alpha-3 catenin	10q21.3	RV/BiV	
	*DES*	Desmin	2q35	LV/BiV	
	*FLNC*	Filamin C	7q32.1	LV	
	*LDB3*	LIM domain binding 3	10q23.2	RV	
	*LMNA*	Lamin A/C	1q22	LV/BiV	
	*TEMEM43*	Transmembrane protein 43	3p25.1	RV/BiV	
	*TGFB3*	Transforming growth factor beta-3	14q24.3	RV	
	*TJP1*	Tight junction protein 1	15q13.1	RV/BiV	
	*TTN*	Titin	2q31.2	RV/LV/BiV	
	*RBM20*	RNA-binding motif protein 20	10q25.2	LV	
**Sodium transport**	*SCN5A*	Sodium voltage-gated channel alpha subunit 5	3p22.2	RV/LV/BiV	
**Calcium homeostasis**	*PLN*	Phospholamban	6q22.31	LV/BiV	

RV: right ventricular involvement; LV: left ventricular involvement; BiV: biventricular involvement.

**Table 2 biomedicines-10-02619-t002:** Main findings supporting the role of the Wnt/β-catenin pathway in different genetic scenarios.

Gene	Model Used	Main Finding	Wnt Alteration
* **PKP2** *	hiPSC-CM [14]	*PKP2* mutated is associated with increased lipogenesis	↓ β-catenin in nucleus
	hiPSCs-CM from ACM patients with *PKP2* mutations [16]	PG translocates into the nucleus and competes with β-catenin	↓ β-catenin activity
	*PKP2* knock-down HL-1 [15]	Activation of the Hippo pathway	↓ Transcription of canonical Wnt target genes
* **DSP** *	*DSP*^±^ mice [11]	Adipocytes in ACM originate from second heart field cardiac progenitors	Suppressed expression of c-Myc
	FAPs from *DSP*^±^ mice [13]	FAPs express desmosome proteins and differentiate into adipocytes through Wnt pathway-dependent mechanisms.	↓ expression of Wnt target genes
* **DSP** *	dsp knock-down *Danio rerio* (zebrafish) [24]	Wnt, TGFβ, and Hippo pathways were deregulated	Wnt pathway was the most affected pathway
* **JUP** *	Mouse model overexpressing truncated *JUP* [13]	Nuclear PG is essential for differentiation into adipocytes	↓ Wnt target genes
* **DSG** *	*DSG* mut mice, *Danio rerio* (zebrafish) and ACM patients [25]	Reduced expression of *Lgals3/GAL* genes	*Lgals3/GAL* regulate Wnt, Hippo, and TGFβ pathways

CM: cardiomyocyte; FAP: fibroadipose progenitor; hiPSC: human induced pluripotent stem cell; PG: plakoglobin; PKP2: plakophilin-2; DSP: desmoplakin; GAL: galectin-3; ACM: arrhythmogenic cardiomyopathy. Arrows denote trend.

**Table 3 biomedicines-10-02619-t003:** Online databases containing information about ncRNAs.

	Database	ncRNA Type	Year	N° ncRNAs	Species
**General**	NCBI	All	2021		From virus to Hsa
	RNACentral [83]		2021	±18,000,000	From virus to Hsa
	RFam [84]		2021	±14,800	From virus to Hsa
**Small** **ncRNA**	miRbase [85]	miRNA	2019	±38,600	271 species
	Diana-TarBase [86]	miRNA	2018	±1,000,000	18 species
	CircBase [87]	cicRNA	2014	±50,000	6 species
	CircInteractome [88]	cirRNA	2016	±117,000	Hsa
	Snopy [89]	snoRNA	2013	±13,800	41 species
**lncRNA**	Noncode [90]	lncRNA	2021	±645,000	Hsa, Mmu
	Diana-LNBase [91]		2020	±500,000	19 animals, 23 plants

Hsa: Homo sapiens; Mmu: Mus musculus.

**Table 4 biomedicines-10-02619-t004:** Main competing endogenous RNAs (ceRNAs) studied in cancer research that regulate the most affected pathways involved in arrhythmogenic cardiomyopathy.

	Direct Target	Indirect Target	Mechanism of Action	Potential Effect on ACM Pathogenesis
**MALAT1**	miR-145 [94]	TGFB	miRNA sponge	Fibrosis
	miR-382-3p [95]	Resistin	miRNA sponge	Adipogenesis
**HOTAIR**	miR-34a [96]	Sirtuin 1	miRNA sponge	Adipogenesis
	miR-613 [67]	Connexin-43	miRNA sponge	Conexome structure
**MIAT**	miR-24 [66]	TGFB1, Furin	miRNA sponge	Fibrosis
**KCNQ1OT1**	miR-214-3p [97]	TGFB1	miRNA sponge	Fibrosis
**MRLP23-AS1**	miR-30b [98]	β-catenin	miRNA sponge	Wnt pathway
**lincRNA-p21**	β-catenin (mRNA) [99]		Inhibition of translation	Wnt pathway
**LSINCT5**	miR-30a [100]	β-catenin	miRNA sponge	Wnt pathway
**LINC00467**	Binding EZH2 [101]	DKK1	Regulation of transcription by epigenetic mechanisms	Wnt pathway
**DANCR**	β-catenin (mRNA) [102,103]		Inhibition of translation	Wnt pathway
	miR-216a [104]	β-catenin	Inhibition of translation	Wnt pathway
	miR-214, -320a [102,103]	β-catenin	Inhibition of translation	Wnt pathway
**UCA1**	miR-18a [105]	YAP1	miRNA sponge	Hippo pathway
**FLVCR1-AS1**	miR-513 [106]	YAP1	miRNA sponge	Hippo pathway
**RP11-323N12.5**	binding C-Myc to YAP1 promoter [107]		Transcription of YAP1 mRNA	Hippo pathway
**circYap**	YAP1 mRNA and translation protein complex [108]		Inhibition of translation	Hippo pathway
**circRNA_010567**	miR141 [109]	TGFB1	miRNA sponge	Fibrosis
**circAXIN1**	[77]		Translate into Axin protein named AXIN1-29545aa	Wnt pathway

DANCR: Differentiation antagonizing non-protein coding RNA; HOTAIR: lncRNA-HOX transcript antisense RNA; MALAT1: lncRNA metastasis-associated lung adenocarcinoma transcript 1; MIAT: myocardial infarction associated transcript; UCA1: urothelial cancer associated 1.

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
