# Peer review of "Non Coding RNAs as Regulators of Wnt/β-Catenin and Hippo Pathways in Arrhythmogenic Cardiomyopathy"

_biomedicines, 2022, doi:10.3390/biomedicines10102619_

Round 1

Reviewer 1 Report

The review "Non coding RNAs as regulators of Wnt/β-catenin and Hippo pathways in Arrhythmogenic cardiomyopathy" by Piquer-Gil and colleagues, is a comprehensive, well written and nicely illustrated summary on current literature concerning Wnt/β-catenin and Hippo/YAP-TAZ signaling regulation and cross-talk in Arrhythmogenic cardiomyopathy.

Minor points:

Line 37: the DSC2 gene name should be in italics.

Line 170: beside the "junk" term, I would also suggest mentioning the alternative term "spacer".

For instance: (...) previously named "junk DNA" and now more precisely termed "spacer DNA". (...)

Figure 4: are the authors sure that "SnRNA" stands for "Small nucleotide RNA" and not for "Small nuclear RNA"?

Line 198: Does "lenguaje" stand for "language"?

Line 291: I would suggest concluding the sentence with "(...) final function and so on", instead of "...etc".

Line 301 "Thanks" is not very appropriate for the scientific language. I would suggest more formal synonyms. For instance: "Due to their ability ..."

Figure 7: A space is missing between "c)" and "RNA stabilization"

Line 339: same comment as for line 301.

Line 395 (Table 3, note 1): the species name for humans is "Homo sapiens" written in italics and with the adjective "sapiens" all lowercase (Linnaeus, 1758). The species name for the mouse is "Mus musculus" written in italics and with the "M" of "Mus" in uppercase (Linnaeus, 1758)

Line 401: as a matter of taste, I would avoid the term "Fortunately", writing instead "However, due to the importance of both pathways (...)"

Line 412: lncRNAs (pluriel)

Line 418: I would suggest "Yang et al.", or "Yang and colleagues", replacing "Yang et col"

Table 4: Please check upper/lowercase in "Direct target" and "Indirect Target"

Line 438; "increase" or increases"? Or maybe "its nuclear localization increases"?

Line 458: "Task force or Padua" or "Task force of Padua"?

Line 470: "interacctions" should be "interactions"

Major points

The review seems a bit "mammalian-centric" in evaluating experimental systems for AC (or ACM) modeling and pathway dysregulation.

By interrogating PubMed with the keywords "Arrhythmogenic Wnt Hippo" 12 articles are found. By excluding those that are Reviews (6), 6 Research Articles remain (see below), reporting scientific evidences (researches) on Wnt and Hippo involvement in AC (or ACM), that could be considered by the authors to provide a more complete analysis of the recent literature (in the last 5 years) related to this topic.

By further excluding articles already cited by the authors (Rouhi et al., Chen et al., Zhang et al.,) the following three articles remain, for their consideration. Of note, two of them (Cason et al.; Giuliodori et al.) corroborate Wnt - Hippo pathway involvement within the Vertebrate group (evidences from the Danio rerio species, a popular animal model in AC), supporting the conservation of this multi-pathway signature during evolution, and promoting the use of simplified animal models as promising platforms to investigate pathway regulation and crosstalk in AC.

--------------

PubMed search with keywords: Arrhythmogenic Wnt Hippo (12 results in total; 6 results represented by research articles, after excluding 6 reviews)

Not cited by the authors:

Novel pathogenic role for galectin-3 in early disease stages of arrhythmogeniccardiomyopathy.

Cason M, Celeghin R, Marinas MB, Beffagna G, Della Barbera M, Rizzo S, Remme CA, Bezzina CR, Tiso N, Bauce B, Thiene G, Basso C, Pilichou K. Heart Rhythm. 2021 Aug;18(8):1394-1403. doi: 10.1016/j.hrthm.2021.04.006. Epub 2021 Apr 19.PMID: 33857645 

Isolated left ventricular arrhythmogenic cardiomyopathy: A case report.

Barbosa-Barros R, Pérez-Riera AR, de Abreu LC, de Sousa-Rocha RP, Oliveira da Costa Lino D, Baranchuk A, Zhang L.J Electrocardiol. 2017 Jan-Feb;50(1):144-147. doi: 10.1016/j.jelectrocard.2016.09.003. Epub 2016 Sep 8.PMID: 27742062

Loss of cardiac Wnt/β-catenin signalling in desmoplakin-deficient AC8 zebrafish models is rescuable by genetic and pharmacological intervention.

Giuliodori A, Beffagna G, Marchetto G, Fornetto C, Vanzi F, Toppo S, Facchinello N, Santimaria M, Vettori A, Rizzo S, Della Barbera M, Pilichou K, Argenton F, Thiene G, Tiso N, Basso C.Cardiovasc Res. 2018 Jul 1;114(8):1082-1097. doi: 10.1093/cvr/cvy057.PMID: 29522173

----------------

Already cited by the authors:

The EP300/TP53 pathway, a suppressor of the Hippo and canonical WNTpathways, is activated in human hearts with arrhythmogenic cardiomyopathy in the absence of overt heart failure.

Rouhi L, Fan S, Cheedipudi SM, Braza-Boïls A, Molina MS, Yao Y, Robertson MJ, Coarfa C, Gimeno JR, Molina P, Gurha P, Zorio E, Marian AJ.Cardiovasc Res. 2022 May 6;118(6):1466-1478. doi: 10.1093/cvr/cvab197.

The hippo pathway is activated and is a causal mechanism for adipogenesis in arrhythmogenic cardiomyopathy.

Chen SN, Gurha P, Lombardi R, Ruggiero A, Willerson JT, Marian AJ.Circ Res. 2014 Jan 31;114(3):454-68. doi: 10.1161/CIRCRESAHA.114.302810. Epub 2013 Nov 25.PMID: 24276085 

Profiling of differentially expressed microRNAs in arrhythmogenic right ventricular cardiomyopathy.

Zhang H, Liu S, Dong T, Yang J, Xie Y, Wu Y, Kang K, Hu S, Gou D, Wei Y.Sci Rep. 2016 Jun 16;6:28101. doi: 10.1038/srep28101.PMID: 27307080 

-------------

Reviewer 2 Report

The article entitled: “Non-coding RNAs as regulators of Wnt/β-catenin and Hippo pathways in Arrhythmogenic cardiomyopathy” represents a comprehensive review article describing the role of non-coding RNAs in the regulation of Wnt and Hippo pathways in ACM.

The entire article is well written, nicely illustrated, and accompanied by an appropriate and up-to-date reference list.

The acceptance of the manuscript is highly recommended.

Author Response

We would like to thank the Reviewer 2 for her/his nice comments. 
